# MIRRA: A Modular and Cost-Effective Microclimate Monitoring System for Real-Time Remote Applications

**DOI:** 10.3390/s21134615

**Published:** 2021-07-05

**Authors:** Olivier Pieters, Emiel Deprost, Jonas Van Der Donckt, Lore Brosens, Pieter Sanczuk, Pieter Vangansbeke, Tom De Swaef, Pieter De Frenne, Francis wyffels

**Affiliations:** 1IDLab-AIRO—Ghent University—imec, Technologiepark-Zwijnaarde 126, 9052 Zwijnaarde, Belgium; francis.wyffels@ugent.be; 2Plant Sciences Unit, Flanders Research Institute for Agriculture, Fisheries and Food (ILVO), Caritasstraat 39, 9090 Melle, Belgium; tom.deswaef@ilvo.vlaanderen.be; 3IDLab-MEDIA—Ghent University—imec, Technologiepark-Zwijnaarde 126, 9052 Zwijnaarde, Belgium; Emiel.Deprost@ugent.be (E.D.); Jonvdrdo.VanDerDonckt@ugent.be (J.V.D.D.); 4Department of Industrial Systems Engineering and Product Design, Ghent University, Graaf Karel de Goedelaan 5, 8500 Kortrijk, Belgium; Lore.Brosens@ugent.be; 5Forest & Nature Lab, Ghent University, Geraardsbergsesteenweg 267, 9090 Gontrode, Belgium; Pieter.Sanczuk@ugent.be (P.S.); Pieter.Vangansbeke@UGent.be (P.V.); Pieter.DeFrenne@UGent.be (P.D.F.)

**Keywords:** microclimate, real-time, data acquisition, sensor platform

## Abstract

Monitoring climate change, and its impacts on ecological, agricultural, and other societal systems, is often based on temperature data derived from official weather stations. Yet, these data do not capture most microclimates, influenced by soil, vegetation and topography, operating at spatial scales relevant to the majority of organisms on Earth. Detecting and attributing climate change impacts with confidence and certainty will only be possible by a better quantification of temperature changes in forests, croplands, mountains, shrublands, and other remote habitats. There is an urgent need for a novel, miniature and simple device filling the gap between low-cost devices with manual data download (no instantaneous data) and high-end, expensive weather stations with real-time data access. Here, we develop an integrative real-time monitoring system for microclimate measurements: MIRRA (Microclimate Instrument for Real-time Remote Applications) to tackle this problem. The goal of this platform is the design of a miniature and simple instrument for near instantaneous, long-term and remote measurements of microclimates. To that end, we optimised power consumption and transfer data using a cellular uplink. MIRRA is modular, enabling the use of different sensors (e.g., air and soil temperature, soil moisture and radiation) depending upon the application, and uses an innovative node system highly suitable for remote locations. Data from separate sensor modules are wirelessly sent to a gateway, thus avoiding the drawbacks of cables. With this sensor technology for the long-term, low-cost, real-time and remote sensing of microclimates, we lay the foundation and open a wide range of possibilities to map microclimates in different ecosystems, feeding a next generation of models. MIRRA is, however, not limited to microclimate monitoring thanks to its modular and wireless design. Within limits, it is suitable or any application requiring real-time data logging of power-efficient sensors over long periods of time. We compare the performance of this system to a reference system in real-world conditions in the field, indicating excellent correlation with data collected by established data loggers. This proof-of-concept forms an important foundation to creating the next version of MIRRA, fit for large scale deployment and possible commercialisation. In conclusion, we developed a novel wireless cost-effective sensor system for microclimates.

## 1. Introduction

Climate change is having profound impacts on ecological, agricultural and other societal systems [1,2,3]. Quantifying climate change and its consequences has become a major focus of the physical and life sciences. Our knowledge of the ongoing changes in the climate system, however, predominantly relies on a global network of official weather stations [3,4,5]. Generally, sensors are established according to the guidelines of the World Meteorological Organisation. As a result, data is solely collected in open landscapes 1.25 m to 2 m above short grasses [6] (often on airports) where the wind mixes the air, well away from tall vegetation such as crops or trees [7]. While meteorologists strive to remove what they consider as local “noise” in the data, this “noise”, the local temperature near the ground or below vegetation, is highly relevant for many terrestrial organisms. Indeed, many organisms do not experience climatic conditions in an open field at a height of 1.25 m to 2 m, but occur near ground-level in heterogeneous environments, such as shrublands, mountains and forests [8]. These localised conditions dictate crucial ecosystem processes such as plant and crop growth, and hydrological, nutrient, and carbon cycles. In forests, for instance, tree canopies buffer the microclimate (the local climatic conditions near the ground) by up to 10 ${}^{\circ }C$ from the macroclimate (the climate of a large geographic region) [9]. The existing global network of standardized weather stations is thus insufficient to quantify climate as experienced by most organisms. Moreover, this can form a major threat in combating the effects of climate change on biodiversity, because it is difficult to design policies if measurement data are lacking [5,10,11].

The microclimate is per the definition determined by the vertical, horizontal and temporal complexity of ecosystems, all of which make microclimate predictions from standard weather stations very complex [12]. Quantifying the true amount of climate change experienced by organisms and its impacts on ecosystems is only possible when taking the microclimate into account. To achieve a thorough assessment of its importance, a first step is to robustly and accurately quantify microclimate itself. Data loggers must collect a suite of climatic conditions including temperature, air humidity, light availability and soil moisture. Real-time readout is desirable, since manual readout is costly and malfunctioning nodes can be detected and replaced faster. As a result, wireless sensor technologies are excellent candidates for microclimate monitoring [13].

Presently, microclimate monitoring stations can roughly be divided in two categories. First, miniature loggers (e.g., LASCAR, HOBO, TMS [14], or iButtons) are simple, small yet robust, and low-cost devices with a per unit cost of EUR 30 to EUR 80 [15,16]. These measure only a limited set of variables, typically temperature and air humidity. Manual data readout via regular site visits is required though in most cases, which results in significant additional costs due to transport, accommodation and working hours. Second, high-end comprehensive weather stations exist, which can simultaneously measure air temperature, relative humidity, radiation, barometric pressure and vapour pressure and report these in real real-time via a 3G or 4G uplink [17,18]. While these are excellently suited for microclimate measurements, they are expensive, with an approximate price per unit of EUR 1000 to EUR 2000, depending on cable length and number of variables measured [19]. In addition, they are generally energy demanding, thus requiring an external power supply or solar panels, increasing overall costs and reducing flexibility. For instance, in forests, it is near impossible to collect sufficient solar energy at ground level. Moreover, sensors are connected using cables, which is often limited to distances of 50 m and a costly aspect for analogue sensors. In summary, microclimate science needs a low-cost, modular, energy-efficient and real-time monitoring system.

Here, we propose a modular system: MIRRA (Microclimate Instrument for Real-time Remote Applications). It aims to tackle key challenges in microclimate, while remaining a low-cost system. The primary goals of MIRRA are to provide a flexible, low-cost and online platform that fills the gap between offline data loggers and expensive online data loggers highlighted in the previous paragraph. MIRRA is a wireless solution, thus simplifying interconnectivity that uses modular nodes. Appropriate analogue and digital sensors are connected to each node, thus increasing the flexibility of the system for various applications. Data can be uploaded to a server in near real-time, reducing costs of readout and avoiding data loss in case the device is lost, stolen or damaged [5].

First, we discuss the technical aspects of the MIRRA system, including inter node communication and data retrieval in Section 2. Then, we set-up an experiment to evaluate the performance of the system and illustrate the effects of microclimate in Section 3. In Section 4, we analyse the data and in Section 5, we make suggestions for future improvements.

## 2. MIRRA System Operation, Architecture and Performance

In this section, we detail the operating principles and performance of MIRRA, including communication, power consumption, and user interfacing. The entire system is open source and can be constructed and programmed from the design and code files, hosted on GitHub (https://github.com/opieters/MIRRA, accessed on 7 June 2021) under a permissive Apache 2.0 licence.

### 2.1. Network Architecture

MIRRA employs a star network, depicted in Figure 1. A central node, henceforth referred to as gateway, collects the data directly from each sensor node. This design choice simplifies the functionality of nodes and reduces power-requirement of nodes since they do not need to forward messages generated by other nodes. The gateway collects data from all assigned nodes and uploads this data to a server.

Nodes connect to the gateway using a Long Range (LoRa) transceiver. LoRa is a low-power technology that is optimised for long-range optimisation thanks to the usage of sub-gigahertz frequencies, developed by Semtech Corporation (Camarillo, CA, USA) [20,21]. In this work, we only rely on simple transmit/receive functionality of the LoRa PHY modulation and do not use the LoRaWAN networking to reduce the overall complexity. Bidirectional communication is implemented to ensure the data is correctly received by the gateway before deletion in the sensor node. To further optimise power consumption, time synchronisation is used. Each node is equipped with a Real-Time Clock (RTC) to awake the system when a communication event occurs and/or sensor data has to be collected. Measurements are performed by all nodes at the same time point, but each node is sequentially contacted by the gateway. Additionally, sensor data from multiple time points is aggregated in a single communication event to reduce power consumption.

The gateway stores all sensor data and uploads it to a central server. Frequent updates are possible, at the expense of increased power consumption, so it is advised to aggregate sensor data before uploading. Data can be uploaded using a 2G (Second Generation) or WiFi (Wireless Fidelity) uplink. A WiFi uplink is convenient and cost-saving if the gateway can be powered from a central supply. For remote operation, the 2G uplink is appropriate. The data is uploaded using the Message Queuing Telemetry Transport (MQTT) protocol.

In our tests, we kept the distance between the sensor node and gateway limited to 100 m. This range can be extended using a suitable antenna to multiple kilometres in urban areas and even 15 km in rural areas [22]. The actual range depends on a large number of time- and site-specific conditions such as the weather, surrounding buildings and/or terrain variations.

### 2.2. Hardware Description, Operating Specifications

An overview of the node’s technical data is provided in Table 1. MIRRA is designed with microclimate monitoring in mind, even in challenging conditions such as arctic tundra, desert and rainforests. Loggers are able to withstand temperatures as low as −30 ${}^{\circ }C$, and as high as 85 ${}^{\circ }C$. Moreover, MIRRA consumes very little power between measurements. A large Lithium Thionyl Chloride battery of 2.6 A
h ensures long on-time without battery replacement. This battery cannot be charged, but if it is drained, it can be replaced by the user. This battery type was preferred over other technologies such as Lithium-Ion (Li-Ion) batteries because of its good performance in cold environments [23]. Actual lifetime is dependent on the employed sensors, measurement frequency and communication interval.

The gateway has a 2G communication module for the data upload. Because this technology was not optimised for low-power operation, it cannot be powered from the main battery due to too high current peaks. Therefore, a separate 10,000 mAh Li-Ion battery is added. Consequently, communication might be interrupted when the temperature drops below 0 ${}^{\circ }C$ (without data loss). In the future, more power-efficient technologies such as NB-IoT (Narrow-Band Internet of Things) or LTE-M (Long Temp Evolution Machine Type Communication) are possible alternatives. Table 2 summaries the most important characteristics for battery lifetime computations. Adding additional Lithium Thionyl Chloride batteries to replace the Li-Ion battery was not preferred because this would significantly increase the cost of the gateway.

For a system with a one gateway and one sensor node, we estimate from Table 1 and Table 2 that the current battery life for a sensor node is 434 days. For the gateway, the main battery lifetime is approximately 1552 days (using a single 2.6 A
h battery). The calculation is detailed in Appendix A and assumes that three sensors are read out every 20 min. Data are uploaded to the gateway and server on a daily basis. The conditions for this calculation are the same as in the trial (Section 3). The lifetime of the Li-Ion-battery is approximately 3 years and 6 months (full capacity available). However, the effective capacity of Li-Ion batteries degrades dramatically with decreasing temperature [24]. As a result, the effective lifetime will be lower and dependent on the weather conditions.

While most experimental designs require air and soil temperature, and relative air humidity logging, additional sensors are sometimes employed to measure specific variables. Examples are wind direction and speed, precipitation, soil moisture content and light intensity [3]. To optimise flexibility, MIRRA has multiple digital and analogue sensor ports, including three I^2^C (Inter-Integrated Circuit), two SPI (Serial Peripheral Interface), two one-wire and two analogue ports. We advise to use digital sensors, since the internal ADC (Analogue-to-Digital Converter) has low performance for high and low voltages. Moreover, digital sensors can also use longer cables without reduced sensitivity. Libraries are implemented for several popular temperature and humidity loggers (e.g., DS18B20, SHT3x).

Sensor data is initially stored in the internal memory of each node. When the gateway is offline, sensor nodes store the sensor values in their 4 MB flash memory (up to 210,000 sample points for three sensor readouts per time point). Similarly, the gateway retains the data if the central server is offline. As a result, MIRRA can also be deployed without a gateway in situations where offline monitoring of a single location is sufficient and setups without internet uplink.

Due to the need in many experiments to monitor incident light, air temperature and relative humidity, a custom Printed Circuit Board (PCB) and radiation shield were designed. These can easily be interconnected to one of the I^2^C ports. The large and open design is optimised for maximum passive air flow. An illustration of a sensor node is depicted in Figure 2. A Photosynthetically Active Radiation (PAR) sensor, was not used because it would violate the low-cost requirement. To the best of our knowledge, there is no cost-effective PAR sensor available. As a result, we used an alternative sensor (APDS9306, Broadcom Inc., San Jose, CA, USA) that reports light intensity in lux. It is sensitive to wavelengths from 470 nm to 634 nm (normalised response higher than 0.1) [25].

The light sensor in Figure 2 is covered with a uniform density filter that passes 25% of the available light. This filter was added because in summer, the sensor might otherwise saturate if the gain is incorrectly configured. Direct sunlight corresponds to an intensity of 100,000 lx to 120,000 lx [26], which can cross the maximum value readout value of 1,048,575 (20-bit configuration) when multiplied with the maximal analogue gain value of 18.

### 2.3. Web Interface

Data are collected on a central server into a database. A web interface is also designed to assign nodes to gateways and experiments. This makes it easy to add nodes to existing setups and monitor ongoing experiments. Data can be downloaded as Comma-Separated Values file (CSV-file). For security reasons, it is currently not possible to alter gateway or node settings remotely. The main web interface is depicted in Figure 3.

## 3. Experimental Validation

To evaluate the long-term performance of the system, an experiment was set-up in the “Aelmoeseneie” forest in Gontrode, Belgium (50.98° N, 3.80° E). Environmental conditions were captured using a single sensor node and gateway for one month and compared to reference data loggers to assess performance and the effects of climate change. Table 3 summarises the sensors employed in this trial and their accuracy. Measurement frequency was fixed to 20 min. The soil temperature was monitored 5 cm below ground level. The distance between sensor node and gateway was approximately 2 m. Air temperature and relative humidity were captured at a height of 80 cm in the case of the MIRRA sensor node. The sensors in the MIRRA sensor node were factory calibrated and the collected data was not altered.

MIRRA sensor data were compared to reference data from two commercial data loggers. The first one, hereafter referred to as forest logger, was located in the same forest, at a distance of approximately 50 m. A high- end commercial data logger (CR1000, Campbell Scientific, Logan, UT, USA) was used. This logger was used to compare MIRRA with a high-end commercial system in a micro-climate setting. Only air and soil temperature data were available, using thermocouples (see Table 3). The air temperature probe was located 20 cm above ground level, while the soil temperature probe measures were 5 cm below ground level. These sensors were calibrated using a temperature bath, as detailed in Govaert et al. [27]. Data for light intensity and relative humidity are not available from this measurement location since it is part of an ongoing long-term trial and it was preferred not to alter the ongoing data collection.

The second data logger was located in a nearby open field at ILVO Plant Campus, Caritasstraat 39, 9090 Melle, Belgium (50.99° N, 3.79° E, at a distance of 2.3 km), hereafter referred to as open field logger. The air temperature and relative humidity probe was located 1.4 m above ground level. Data from this data logger was used to highlight the effects of microclimate. The setup is also part of another trial and was not altered for this trial. The sensors employed by each of these data loggers are also listed in Table 3. Due to a defect sensor, there is no data for the air temperature and relative humidity before 26 November 2020.

Figure 4 and Figure 5 visualise the sensor data from the trial as well as the data from the reference loggers. Note that in Figure 4a and Figure 5a, two different y-axis are used. This is needed because light is measured using two distinct quantities: in lx for the MIRRA node, and in W m^−2^ for the open field logger.

## 4. Discussion

The data is correctly transmitted from sensor node to server though the gateway since there is no missing data. As expected, we observe a daily temperature increase during daylight, followed by a drop at night in addition to the variation due to variable weather. For relative humidity, the inverse is observed. The time between the different sources is also well synchronised.

The light intensity peaks and solar radiation are synchronised (no time-offset) in Figure 4a, but the peak intensities differ substantially. The Pearson correlation ρ between both is 0.761. The strong difference on some days such as 6 December and 10 December can be attributed to a different measurement technique: the solar radiation sensor integrates over the measurement period, producing an average radiation figure every 20 min. The light sensor in the MIRRA node, however, measures the instantaneous intensity only. This is necessary to minimise power consumption.

Figure 4b,d show excellent correlation between measured characteristics and the forest reference data for the soil and air temperature. The Pearson correlation coefficients (ρ) of the air temperature and soil temperatures are 0.997 and 0.983, respectively. An offset of 1.6${}^{\circ }C$ is observed for the soil temperature, which can be attributed to a measurement error inherent to the sensor since the factory calibration was used, or the local differences (e.g., in terms of soil moisture) between sites of the forest reference logger and MIRRA sensor node. A linear correction can correct for the offset.

The correlation with the open field reference location for soil and air temperature show similar observations. The correlation coefficient is 0.983 and 0.976 for the air and soil temperature, respectively. These values are slightly lower, as expected. In the open field, the temperature is buffered less, as indicated by the stronger temperature peaks in Figure 4b [28].

The open design of the sensor shield, optimising passive airflow, did not result in saturated relative humidity values (Figure 4c), commonly observed in forests, where the relative humidity is high [29]. The relative humidity is well correlated with the open field reference data (ρ=0.928), though some clear differences are also observable due to the different measurement location. During the day, the relative humidity drops more quickly in the open field compared to the forest location of the node. This is clearly visible on several days, for instance, 28 November, 4 December, 8 December and 10 December. Moreover, the open field soil temperature is also more variable than in the forest conditions (see also Table 4).

Table 4 lists key statistical properties such as the main, standard deviation (σ), minimal, maximal, average of the minimum per day, average of the maximum per day, average of the minimum per night, and average of the maximum per night. The day is defined as the middle twelve hours of the day, and night the first and last six hours of the day.

Correlations of different environmental conditions between the sites is quite high, as indicated by the correlation coefficients and statistical data in Table 4. These are to be expected given the season (winter) and limited duration of the trial. Stronger differences are to be expected during summer when the canopy is closed [9,30]. Moreover, no weather extremes were observed during the trial (very high or low temperatures).

Table 5 indicates the cost associated with building the necessary hardware for MIRRA. Clearly, MIRRA is much more cost-effective when compared to large commercial data loggers at the expense of reduced flexibility. While MIRRA will not replace such data loggers, it can be used complementarily. Compared to cheap offline data loggers, MIRRA is slightly more expensive, but comes with increased functionality and performance. This cost calculation is for low-volume production, so costs will drop when the system is commercialised. Moreover, some components can be eliminated when using a higher degree of integration. For instance, by eliminating the RTC and using a microcontroller module that has a built-in LoRa radio.

MIRRA is primarily designed with microclimate monitoring in forests in mind, but it can also be applied elsewhere. In general, any application where sensing is required over long periods of time with low power consumption and limited data rates is possible. Examples include but are not limited to monitoring of environmental conditions in agricultural fields and mountains, water level measurements in rivers or status monitoring of infrastructure, assuming suitable sensors are connected.

## 5. Future Improvements

MIRRA is currently a proof-of-concept design. It functions properly, as is indicated by the measurements in Figure 4 and Figure 5. Yet, we identified some areas where improvements can further increase performance and reliability.

Power consumption is already very limited, but it can be improved further. In the next version, we aim to work towards an overall lifetime of 10 years for a sample period of 20 min with similar sensors as in this experiment (Table 3). To achieve this, we aim to select more power efficient communication technologies and/or communication settings, and optimise the software to maximise the time in deep sleep.

For now, the code has to be recompiled when one wants to change the sample frequency, connected sensors and/or upload rate. While this mitigates attacks on the nodes, it requires field-researchers to modify the source code. Ideally, this should be configurable in software. Moreover, remote firmware updates are also desirable, in case of long-term operation.

While a 2G GPRS uplink is still possible in most countries, it is being phased out in some parts of the world [31]. A future version should employ newer technologies with improved power savings such as LTE-M and NB-IoT [32]. Coverage of these technologies is improving globally. However, for very remote locations, multi-hop LoRaWAN will be necessary due to the lack of cellular coverage needed for LTE-M and NB-IoT [32,33]. For instance, in Belgium, universal coverage of NB-IoT is available, while in other countries with very remote regions such as Norway and Australia, it is not universally available [34,35,36]. For more information on multi-hop LoRaWAN, we refer to, e.g., Cotrim and Kleinschmidt [37] as a recent review on the subject. A multi-hop structure forwards the data until a location is reaches with cellular coverage, where a gateway uploads the data to the cloud. Yet, industrial application is very limited.

## 6. Conclusions

The open hard- and software MIRRA system aims to advance remote real-time microclimate data collection. Currently, inexpensive data loggers lack the possibility to monitor environmental conditions remotely. While this is possible with large commercial data loggers, these are expensive, limiting their applicability in large trials with many monitoring locations, while high spatial coverage is crucial to study the microclimate. They often also need a solar panel, limiting their applicability in forests. Moreover, inexpensive data loggers are often limited to measure one to three key variables such as temperature and relative humidity. With MIRRA, we address these issues. MIRRA sensor nodes feature a modular design that allows multiple digital and analogue sensors to be connected. Data are monitored in real-time, and transmitted to a gateway, which acts as intermediary. Data are subsequently uploaded to a server for further processing and analysis. The system performed excellently over the trial and the data was highly comparable to that of two reference locations in a forest and open field in close proximity to the location of the sensor node. Though MIRRA is primarily designed for microclimates, it can also be used in other applications with low-power sensors that have to measure over long periods of time. Improvements are needed to optimise the battery lifetime and Internet connectivity, yet this proof of concept is an important step towards more widespread and real-time microclimate monitoring. It addresses the need in providing increasingly accurate data for research in a cost-effective manner without sacrificing expandability or accuracy.

## Figures and Tables

**Figure 1 sensors-21-04615-f001:**
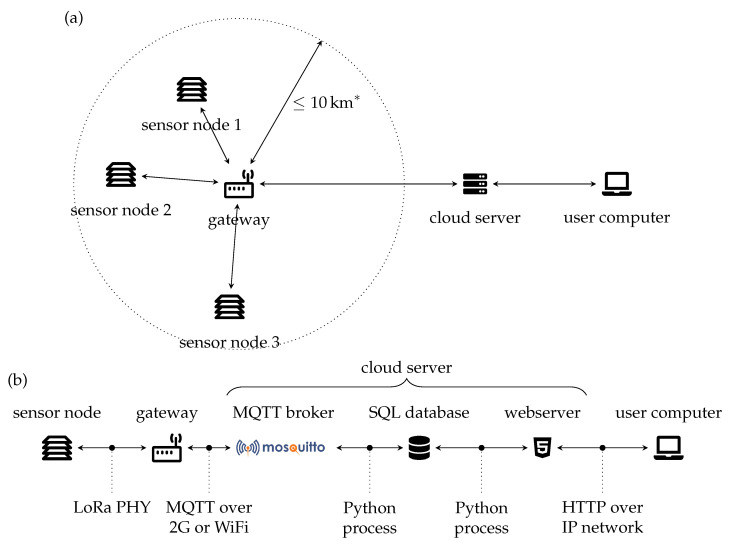
MIRRA network architecture and functional blocks. (**a**) Illustration of how the different components in the system communicate with each other. (*) Theoretical range indicated, actual range dependent on antenna used, location and environmental conditions. Set-up was tested up to 100 m. (**b**) A functional block diagram that illustrates the different (sub)components of MIRRA.

**Figure 2 sensors-21-04615-f002:**
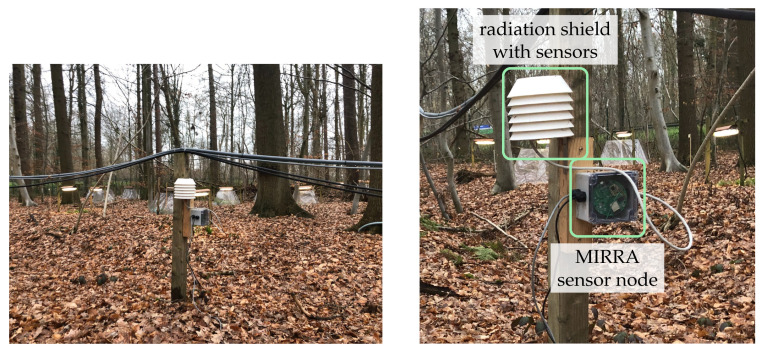
Sensor node in the experimental set-up, consisting of a sensor board (square block), radiation shield (white) and ground probe. This set-up is located in the “Aelmoeseneie” forest in Gontrode, Belgium (50.98° N, 3.80° E).

**Figure 3 sensors-21-04615-f003:**
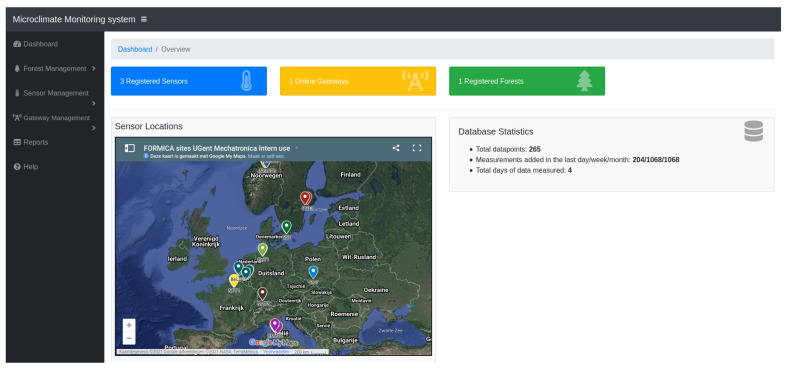
Web interface for node status monitoring and assignation.

**Figure 4 sensors-21-04615-f004:**
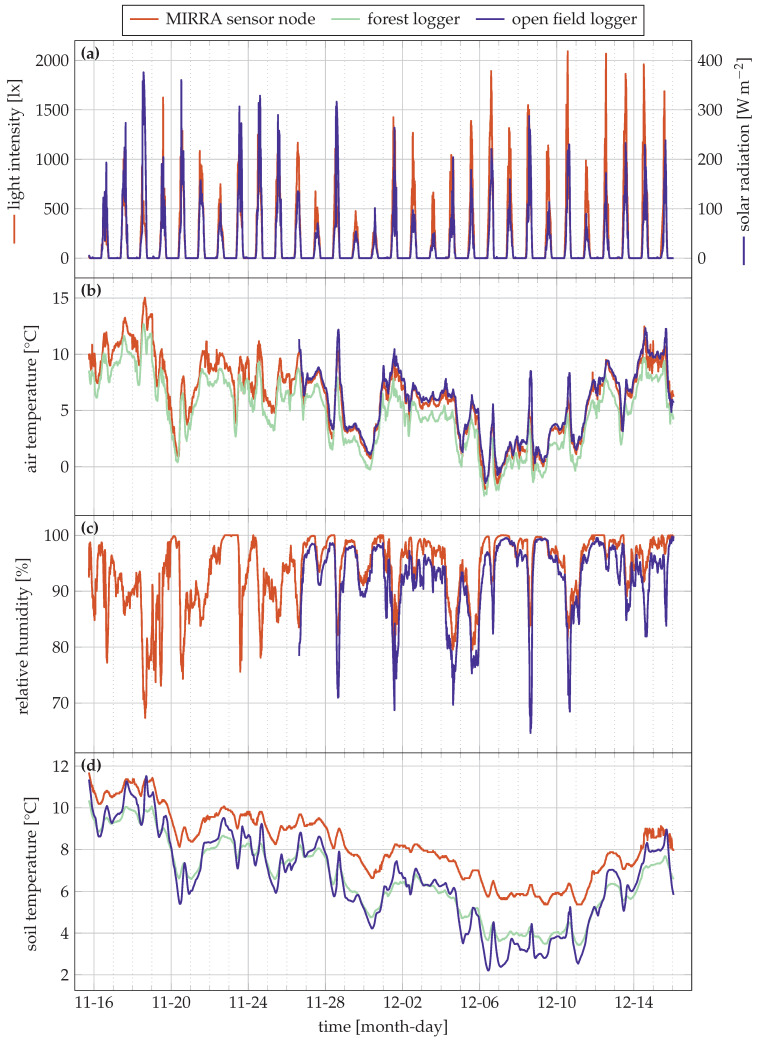
Measurement data of the MIRRA sensor during the month-long trial in 2020 and reference locations. Each of the subfigures details a separate quantity: (**a**) Light intensity in lx (left axis) and W m^−2^ (right axis); (**b**) air temperature; (**c**) relative humidity and (**d**) soil moisture. For the open field logger, there is no data for some quantities before 26 November 2020.

**Figure 5 sensors-21-04615-f005:**
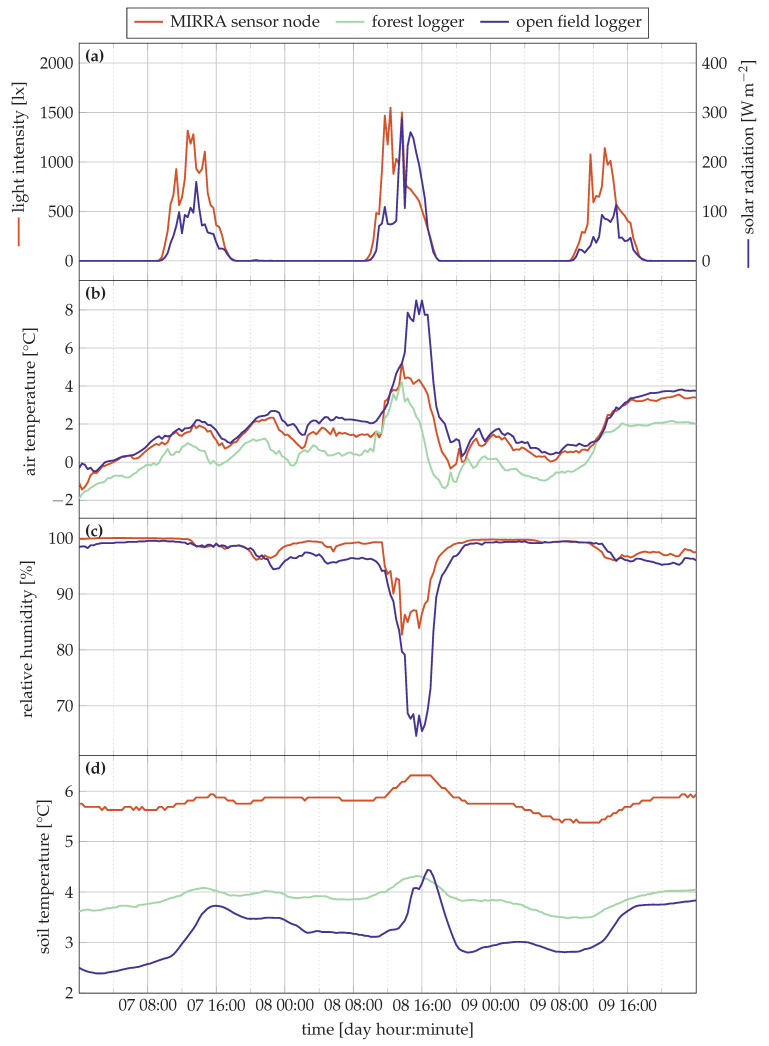
Detailof measurement data between 7 and 10 December 2020. Each of the subfigures details a separate quantity: (**a**) Light intensity in lx (left axis) and W m^−2^ (right axis); (**b**) air temperature; (**c**) relative humidity and (**d**) soil moisture.

**Table 1 sensors-21-04615-t001:** Most important technical specifications of a MIRRA sensor node for field measurements and operating time estimations, measured at an ambient temperature of 20 ${}^{\circ }C$.

Characteristic	Value
microcontroller module	ESP32-WROOM-32
flash memory	4 MB
sRAM memory	540 KiB
deep sleep power consumption	49 μA
normal operation power consumption	30 mA
LoRa transmission	44 mA
LoRa reception	8 mA
typical on-time (3 sensor readouts, no communication)	8 s
typical sensor readout duration in the experiment (3 sensor readouts, approx.)	5 s
LoRa transmission duration (3 sensor readouts, approx.)	1 s
LoRa typical reception duration (approx.)	5 s
internal ADC accuracy	12 bit
battery capacity	2.6 A h

**Table 2 sensors-21-04615-t002:** Most important technical specifications of the MIRRA gateway (measured at an ambient temperature of 20 ${}^{\circ }C$).

Characteristic	Value
microcontroller module	ESP32-WROOM-32
flash memory	4 MB
sRAM memory	540 KiB
deep sleep power consumption	56 μA
normal operation power consumption	35 mA
LoRa transmission	45 mA
LoRa reception	9 mA
2G stand-by	47 mA
2G data communication	475 mA
typical 2G transmission (with 3 sensor nodes)	32 s
typical 2G standby duration (with 3 sensor nodes)	68 s
typical on-time (with 3 sensor nodes) (on transmitting ore receiving)	20 s
typical time LoRa transmission duration (with 3 sensor nodes)	3 s
typical time LoRa reception duration (with 3 sensor nodes)	40 s
battery capacity	2.6Ah or 5.2 A h

**Table 3 sensors-21-04615-t003:** Sensor overview in the trial. Accuracies are as specified by the manufacturer in the datasheet. For analogue sensors, the accuracy cannot always be included because it is determined by a combination of the sensor accuracy and sensitivity, cabling and data logger used.

Sensor	Device Part	Accuracy
MIRRA node air temperature	SHT35 (Sensirion AG, Switzerland)	±0.2 ${}^{\circ }C$
MIRRA node relative humidity	SHT35 (Sensirion AG, Switzerland)	±2%
MIRRA node soil temperature	DS18B20 (Maxim Integrated, San Jose, CA, USA)	±1 ${}^{\circ }C$
MIRRA node light intensity	APDS9306 (Broadcom Inc., San Jose, CA, USA)	±0.1 lx
forest logger air temperature	Type T miniature thermocouple 406-580 (TC Direct, The Netherlands)	analogue
forest logger soil temperature	Type T miniature thermocouple 406-580 (TC Direct, The Netherlands)	analogue
open field logger air temperature	CS215 (Campbell Scientific, Logan, UT, USA)	±0.9 ${}^{\circ }C$
open field logger relative humidity	CS215 (Campbell Scientific, Logan, UT, USA)	±2%
open field logger soil temperature	107 Temperature Probe (Campbell Scientific, Logan, UT, USA)	±0.5 ${}^{\circ }C$
open field logger light radiation	LP02 pyranometer (Campbell Scientific, Logan, UT, USA)	analogue

**Table 4 sensors-21-04615-t004:** Overview of statistical properties of the three data sets. σ is the standard deviation. Light is reported in lx and W m^−2^ s^−1^ for the MIRRA and open field loggers, respectively. (${}^{*}$) these variables are computed only between 26 November and 15 December because no data is available for the open field logger for earlier time points.

Variable	MIRRA	Forest Logger	Open Field Logger
mean air temperature^*^ [°C]	4.91	3.59	5.41
*σ* air temperature^*^ [°C]	2.96	2.76	3.00
min air temperature^*^ [°C]	−1.97	−2.57	−1.45
max air temperature^*^ [°C]	12.47	9.69	12.32
mean min day air temperature^*^ [°C]	3.88	2.48	4.40
mean min night air temperature^*^ [°C]	3.23	1.90	3.68
mean max day air temperature^*^ [°C]	6.93	5.51	7.76
mean max night air temperature^*^ [°C]	6.41	4.71	6.70
mean soil temperature [°C]	8.20	6.56	6.56
*σ* soil temperature [°C]	1.61	1.79	2.18
min soil temperature [°C]	5.38	3.42	2.20
max soil temperature [°C]	11.69	10.35	11.52
mean min day soil temperature [°C]	8.00	6.36	6.08
mean min night soil temperature [°C]	7.92	6.24	5.93
mean max day soil temperature [°C]	8.58	6.95	7.46
mean max night soil temperature [°C]	8.66	7.06	7.41
mean relative humidity^*^ [%]	95.68		92.48
*σ* relative humidity^*^ [%]	4.77		6.52
min relative humidity^*^ [%]	76.86		64.63
max relative humidity^*^ [%]	100.00		99.80
mean min day relative humidity^*^ [%]	89.38		83.20
mean min night relative humidity^*^ [%]	92.50		88.55
mean max day relative humidity^*^ [%]	98.51		95.87
mean max night relative humidity^*^ [%]	99.28		97.69
mean light [lx or W m^−2^ s^−1^]	181.36		27.89
*σ* light [lx or W m^−2^ s^−1^]	343.72		58.25
max light [lx or W m^−2^ s^−1^]	2092.00		375.79

**Table 5 sensors-21-04615-t005:** Cost-calculation of MIRRA hardware. The *other* category includes all remaining components not covered by the other columns, including integrated circuit and passive components, connectors, and batteries.

Module Type	Cost Category [€]								Total Cost [€]
	passive components	LoRa module	microcontroller	RTC	connectors	enclosure	uplink	other	
sensor node	5.53	17.11	3.81	2.57	10.83	22.16		13.09	52.94
radiation shield	0.323					10.2		9.143	
gateway	5.56	17.11	3.81	2.57	10.09	22.16	5.34	68.68	113.16

## Data Availability

The data used in this publication is available on Zenodo (doi:10.5281/zenodo.5069798).

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
