# Peer review of "MIRRA: A Modular and Cost-Effective Microclimate Monitoring System for Real-Time Remote Applications"

_sensors, 2021, doi:10.3390/s21134615_

Round 1

Reviewer 1 Report

The proposed manuscript has presented a real-time remote application for microclimate monitoring. It is an impressive tool and manuscript. The authors have drafted the manuscript describing the tool in a comprehensive manner with most of the possible details. However, it is advised to consider the following comments to improve the revised draft:

  1. A detailed block diagram describing the proposed tool (MIRRA) can be beneficial for the readers and will be efficient to describe the system.

  2. 'Statistical details of the generated dataset' and 'data acquisition systems' are the two points where the authors can discuss further to include details in the manuscript.  

3. The authors must put some details on 'Motivation' and 'Applications' of MIRRA application. Also, similar information must be provided in the Abstract and Conclusion sections.  

4. There are several statements in the manuscript which are very long and missing meanings sometimes. It is advised to revise such parts of the manuscript text.

Lines 10-13: 'The overarching ... connectivity.'

Lines 139-141: 'While most ... light intensity.'

Lines 155-157: 'A Photosynthetically ... requirement.'

Lines 270-271: 'While this ... accuracy.'

Reviewer 2 Report

- The redaction of some sentences could be improved, including but not limited to lines 22, 74, 80, and 131.

- Consider adding the memory and processor specifications to Table 1.

- Line 178, please justify the distance chosen in your experimental setup (2m) when you advertise operational ranges up to 10km (15km in rural areas).

- Line 186, please justify the choice to employ a different measurement height than the reference system (80cm vs 20cm). As shown in Figure 4(b) , this choice had some effect in the measurements.

- The 'forest logger' you employ as reference seems to fulfill a similar role in the study of microclimates as the proposed system. Consider discussing its characteristics and limitations in the introduction section.

- Most interesting to me is that the measurements of the 'open field logger' and your proposed architecture are quite close as evidenced by your p correlation. Some could take this as evidence that the open loggers work well enough for the study of microclimates. Can you visualize some ecosystem where a noticeable difference could be appreciated?

- It could be interesting to discuss the cellular coverage in microclimates of interest. Based on this you could determine if adopting new uplink technologies is a better alternative than improving the multi-hop LoRaWAN capabilities of the system.
